# How Influenza A Virus NS1 Deals with the Ubiquitin System to Evade Innate Immunity

**DOI:** 10.3390/v13112309

**Published:** 2021-11-19

**Authors:** Laurie-Anne Lamotte, Lionel Tafforeau

**Affiliations:** Cell Biology Laboratory, Research Institute for Biosciences, Research Institute for Health Sciences and Technology, University of Mons, 7000 Mons, Belgium; laurie-anne.lamotte@umons.ac.be

**Keywords:** influenza A virus, NS1, ubiquitin, SUMO, ISG, NEDD8, innate immunity, interferon, cytokines, TRIM25, RIG-I

## Abstract

Ubiquitination is a post-translational modification regulating critical cellular processes such as protein degradation, trafficking and signaling pathways, including activation of the innate immune response. Therefore, viruses, and particularly influenza A virus (IAV), have evolved different mechanisms to counteract this system to perform proper infection. Among IAV proteins, the non-structural protein NS1 is shown to be one of the main virulence factors involved in these viral hijackings. NS1 is notably able to inhibit the host’s antiviral response through the perturbation of ubiquitination in different ways, as discussed in this review.

## 1. The Ubiquitin and Ubiquitin-like Systems

### 1.1. The Ubiquitin-Proteasome System

Ubiquitin is a small 8.5 kDa protein consisting of 76 amino acids, expressed in almost every cell type and largely conserved through eukaryotes. The binding of the C-terminal ubiquitin end to the ε-amino group of lysine (K) residue on a protein substrate is called ubiquitination. Ubiquitination is one of the best described post-translational modifications and it regulates, among other things, targeted protein localization, function, cellular trafficking, and degradation by the 26S proteasome [1,2]. Specific substrate ubiquitination requires a three-step enzymatic cascade involving E1, E2 and E3 enzymes. First, E1 enzyme activates the ubiquitin in an ATP-dependent way and transfers it to the ubiquitin-conjugating enzyme, E2. Finally, the ubiquitin ligase E3 catalyzes an isopeptide bond between the ubiquitin molecule and a lysine on the protein substrate [3,4,5]. After this mono-ubiquitination, the ubiquitin moiety itself can be ubiquitinated on eight amino groups: on the ε-amino groups of its seven K residues (K6, K11, K27, K29, K33, K48 and K63) and on the α-amino group of its N-terminal methionine. Various poly-ubiquitin chains can thereby be formed and are decisive for the targeted protein’s fate. The best known poly-ubiquitinations are K48-linked chains, that trigger protein degradation by the 26S proteasome, and K63-linked chains, that control endocytosis, cellular trafficking and protein activity (Figure 1) [2,6,7,8]. The 26S proteasome is a large protein complex with a barrel shape that consists of a catalytic central core (20S) and two 19S regulatory caps at both ends of the core. The 20S core is made of two inner rings of seven β subunits and two outer rings with seven α subunits. The two 19S bind to the 20S and activate it in an ATP-dependent manner, thus allowing the targeted protein to enter the cavity to be proteolyzed into small peptides [9].

Ubiquitinated proteins are subsequently recognized by other proteins or enzymes containing ubiquitin-binding domains, among which are de-ubiquitinase enzymes (DUBs). DUBs are able to hydrolyze isopeptide bonds formed between ubiquitin and the protein substrate or between ubiquitin moieties, with substrate and linkage specificity [10,11]. This allows the recycling of ubiquitin molecules in the cell and other crucial cellular processes that rely on free ubiquitins, such as innate immune response and autophagy (Figure 1) [11,12,13,14].

The number of each enzyme implied in this system reflects their specificity: to date, 9 E1s, 43 E2s, 919 E3s and 126 DUBs have been described in humans [15]. Usually, E2 enzymes define the type of linkage, while E3 ligases determine the substrate specificity. Three main families can be found within E3 enzymes, according to the way E3s trigger ubiquitin transfer from E2 enzyme to the substrate: the really interesting new genes (RING) family, the homologous to E6-associated protein C terminus (HECT) domain family and the RING-between-RING (RBR) family [6,7,16,17,18,19,20,21,22,23]. RING ligases are the most abundant E3 enzymes and act as mediators for direct ubiquitin transfer. They contain a zinc-binding RING domain or a U-box domain, comprising the same fold as RING but lacking zinc-chelating residues. RING ligases can be found in monomeric, homodimeric or heterodimeric states, or can be composed by multiple subunits, as with the cullin-RING ligases (CRLs) [24]. HECT and RBR ligases catalyze ubiquitin transfer through an internal thioester bond before transferring it to the substrate [6,12]. These two differ by their characteristic domain [24]. The whole ubiquitination system, including ubiquitin, enzymes and proteasome, is commonly referred to as the ubiquitin-proteasome system (UPS). Crucially, UPS is a key player in signal transduction pathways, including in innate immune responses. Increasing evidence depicts UPS’s role during viral infections, such as influenza A virus (IAV) infections, notably through tripartite motif assembly domain (TRIM) proteins from the RING family [25,26]. Besides the UPS, there are other systems implying ubiquitin-like and E1, E2 and E3 enzyme-like proteins, such as SUMOylation, ISGylation or NEDDylation [27].

### 1.2. Ubiquitin-like Systems

#### 1.2.1. SUMOylation

As ubiquitination, SUMOylation is a post-translational modification of K residues on a protein substrate by a small ubiquitin-like modifier (SUMO) protein conjugation that can thus compete with K residues ubiquitination. This modification is, for instance, involved in transcription, mRNA synthesis and DNA replication or damage responses [28]. SUMOylation mainly occurs under cellular stress to resolve issue such as hypoxia, heat shock or genotoxic stresses [29]. As the UPS, this system also plays a role in the viral-dependent innate immunity pathway [30], notably by regulating functions of some pattern recognition receptors (PRRs) [31,32] and the stimulation of interferons (IFNs) [33,34].

#### 1.2.2. ISGylation

ISGylation of cellular or viral proteins is also a post-translational modification, induced by type I IFNs stimulation in infected cells [35]. The interferon stimulated gene 15 protein (ISG15) is the first ubiquitin-like protein described in the literature [36,37]. This protein contains two ubiquitin-like domains separated by a linker region [38]. Some studies indicate that ISG15 can be secreted similarly to a soluble cytokine from epithelial cells and lymphocytes to stimulate inflammatory response and IFNγ (type II IFNs) release [39,40,41]. As an ISG protein, ISG15 expression is induced by innate immune responses through type I IFN stimulation [42], but also by other factors [43,44,45]. Similarly, ISG15 E1 (Ube1L/Uba7) [46], E2 (Ube2L6) [47,48] and E3 (Herc5) enzymes are induced by type I IFN expression [49,50,51], as well as its main DUB (USP18) [52]. Herc5 is associated with ribosomes, leading to random and large ISGylations during protein translation. Therefore, several substrates can be ISGylated, even though oligomeric viral structure proteins are mainly targeted by ISG15 to be inactivated [53].

#### 1.2.3. NEDDylation

Conjugation of neural precursor cell expressed developmentally down-regulated protein 8 (NEDD8) to K residues of a protein substrate involves NEDD8-activating enzyme E1 (NAE), NEDD8-conjugating enzyme E2 (Ube2M and Ube2F) and different substrate-specific E3 enzymes (reviewed in [54]). The best characterized substrates of NEDD8 are cullin proteins, components of CRLs. Upon cullin NEDDylation, CRL undergoes conformational change which facilitates ubiquitin transfer from E2 enzyme to the substrate [55]. NEDDylation can be reversed by de-NEDDylase enzymes such as COP9 signalosome (CSN), which thus inactivates CRLs [56,57]. More recently, non-cullin targets of NEDD8 were also highlighted, including the itchy-homolog (ITCH) [58], Parkin [59] and MDM2 [60,61] E3 ligases. NEDDylation of these other substrates seems to enhance their stability [62].

## 2. Influenza A Virus and Ubiquitin

### 2.1. Influenza A Virus

Influenza A virus is a member of the Orthomyxoviridae family and consists of eight single-stranded negative sense RNA (ssRNA(−)) that encode 10 main proteins [63]. Hemagglutinin (HA) and neuraminidase (NA) are surface glycoproteins that facilitate viral entry and release, respectively. The M2 protein is an ion channel found in the virus envelope, itself surrounding a matrix formed by M1 protein oligomers. Each RNA segment is encapsidated by many nucleoprotein monomers (NP) and associated with an RNA-dependent RNA polymerase (PB1, PB2 and PA subunits), forming a viral ribonucleoprotein (vRNP) complex. The genome also encodes a nuclear export protein (NEP or NS2) and a non-structural protein (NS1) [63]. More recently, other non-structural proteins produced from alternative splicing have been identified in infected cells (PB1-F2, PA-X, PA-N40) [64]. The IAV polymerase acts in two ways during the course of infection, as it first triggers viral RNA transcription for proteins expression, and then RNA replication for viral particles production through viral positive-sense complementary RNA (cRNA) production [65,66]. Previous studies suggested that these two successive functions could be regulated by post-translational modifications of viral polymerase proteins such as PB1, PB2, PA and NP phosphorylation [67] and ubiquitination [68] or PB1 and NP SUMOylation [69,70].

### 2.2. Importance of the UPS in the IAV Life Cycle

As mentioned above, UPS has a crucial impact on the IAV life cycle. On the one hand, viral proteins ubiquitination mainly down-regulates their stability and function, but on the other hand, the lack of these modifications is clearly detrimental for proper infection [71,72,73,74,75,76,77,78]. For instance, it was shown that 26S proteasome inhibition prevents IAV entry in the cell and negatively impacts viral polymerase activity [68,74,79], while K48-linked poly-ubiquitination of viral proteins lead to their degradation [72,75]. IAV is no exception, and a lot of viruses therefore evolved to counteract and exploit the UPS. For example, some viral proteins can target cellular proteins for K48-linked degradation or use cellular DUBs to protect themselves, as well as change cellular E3s’ specificity or encode their own ubiquitin ligase enzymes [80,81,82]. Moreover, the UPS is particularly up-regulated in IAV-infected cells [83].

The IAV life cycle follows different steps in which the UPS is involved, beginning with viral entry in host cell [26]. First, the viral particle binds to sialic acids on the host cell surface by its HA protein [84]. This binding leads to the virus’ internalization in endosomes [63], and the viral particle is acidified by M2 ion channel opening [85,86]. Upon low pH, interactions between viral proteins weaken and HA undergoes a conformational change that triggers the fusion of the viral envelope with the endosome membrane [87], leading to the release of the viral genome into the cytosol by M1 destabilization. At these steps, ITCH E3 ligase mediates ubiquitination and degradation of M1, facilitating vRNPs’ release from endosomes [77], and the cullin 3 (Cul3) E3 promotes endosome maturation and thus uncoating process [88,89].

Following this uncoating, vRNPs are then imported into the nucleus through importin α/β and are processed by the viral polymerase and host factors for transcription of viral mRNA and replication [90,91,92]. In the nucleus, the interaction between NP and RNA seems to be regulated by K184 residue mono-ubiquitination of NP. This modification is crucial for viral replication and is reversed by the cellular DUB USP11 [71,73]. NP can also be poly-ubiquitinated on other K residues by other E3 ligases without targeting it for degradation. In fact, all viral polymerase proteins (PB1, PB2, PA and NP) are ubiquitinated and these modifications lead to enhanced polymerase activity and accumulation of viral RNA, cRNA and mRNA in the infected cell [68]. Besides ubiquitination, NP can also be SUMOylated at K7. This modification plays a crucial role in NP intracellular trafficking and therefore ensures its early nuclear retention for mature vRNPs’ assembly. Notably, this SUMOylation is counteracted at the late stage of infection by host caspase-dependent cleavage [70].

Meanwhile, viral proteins are produced in the cytoplasm and on the membrane of the endoplasmic reticulum, and are afterwards assembled into new virus particles with the newly synthetized vRNPs and the host plasma membrane-derived envelope, leading to the budding of new virions from the infected cell [63]. Aminoacyl-tRNA synthase complex-interacting multifunctional protein 2 (AIMP2) is a cellular protein that promotes vRNPs’ nuclear export and replication at the late steps of infection. NEP protects this protein from K48-linked poly-ubiquitination by E3 ligases such as Parkin [93,94]. In turn, AIMP2 inhibits K242 ubiquitination of M1 by ITCH and its subsequent degradation by the 26S proteasome [95]. At the late stages of infection, this K242 residue can therefore be SUMOylated, a crucial modification for vRNPs’ nuclear export [96].

## 3. IFNs and Cytokines Activation: The Innate Immune Response during IAV Infection

### 3.1. IFNs Activation

In an infected cell, foreign and potentially pathogenic materials, namely pathogen-associated molecular patterns (PAMPs), are recognized by pattern recognition receptors (PRRs) localized on the cell and/or endosome membranes or in the cytoplasm. This recognition leads to type I (IFNα et IFNβ) and type III (IFNλ) IFNs as well as pro-inflammatory cytokines production through the activation of signaling pathways [97]. Subsequent paracrine and autocrine signals then establish an antiviral state in the infected cell and in surrounding cells through ISGs expression. Post-translational modifications such as ubiquitination and phosphorylation of several cellular factors regulate this signalization pathway [80,98]. This regulation allows a duration and intensity balance in innate immune response [99]. In fact, excessive type I IFNs production leads to tissue injury and apoptosis induction [100,101].

Among innate immune response regulators, TRIM E3 ligases are key factors, leading to IRF3 and NF-κB activation [102]. A lot of TRIM proteins are known to inhibit IAV through different mechanisms [71,72,103,104,105], and the best example described so far is TRIM25, as an IAV antagonist and innate immunity pathway activator (Figure 2) [106,107]. To ensure an immunity balance, TRIM25 itself can either be K48-linked poly-ubiquitinated to be discarded or de-ubiquitinated by ubiquitin specific protease 15 (USP15) to be stabilized [108]. TRIM proteins possess a conserved architecture consisting of a N-terminal RING domain, followed by one or two B-box domain(s), and a coiled-coil domain (CCD) involved in TRIM dimerization [109,110,111] and in interactions with several proteins [112,113,114,115,116,117]. TRIMs’ C-terminal region is characterized by non-catalytic domains responsible for substrate or other proteins recognition and for subcellular localization [116,118]. Notably, this region possesses a PRY-SPRY domain in half of the TRIM proteins [118].

Retinoic acid-inducible gene I (RIG-I)-like receptors (RLRs) are cytoplasmic PRRs consisting of three described members: RIG-I, melanoma differentiation-associated protein 5 (MDA5) and LGP2. The first and best characterized RLR is RIG-I, which is particularly active during IAV infection and therefore the key sensor in these infected cells [119,120,121]. RIG-I possesses two N-terminally located caspase activation and recruitment domains (CARDs), followed by a central DExD/H-box helicase-like domain with ATPase and translocase activities, a linker (Br) and a C-terminal regulatory/repressor domain (CTR/RD) in its C-terminal region (Figure 3). RIG-I principally detects short 5′-triphosphorylated double-stranded (ds)RNAs (5′-ppp dsRNAs) such as those produced in 5′ and 3′ UTR ends of IAV genome that are hybridized in a panhandle structure [97,121,122,123,124,125]. Therefore, RIG-I senses IAV genomic RNAs associated with NP when they are imported into the nucleus or when they are incorporated into new virion particles at late stages of infection [125,126,127]. The Toll-like receptor 3 (TLR3), a membrane PRR, seems to be also involved in this IAV sensing [128].

In the cytosol of non-infected cells, RIG-I is in an inactive state where its helicase domain binds its CARDs, hiding them from interactions and modifications [125,129,130]. Upon the sensing of a 5′-ppp dsRNA by CTR/RD, RIG-I undergoes conformational changes mediated by ATP hydrolysis [131,132,133]. CARDs become exposed and RIG-I is K63-linked poly-ubiquitinated at K172 residue by TRIM25 [106] and at K849 and K851 residues by the E3 Riplet [134,135,136,137,138], respectively, even though recent studies suggest that RIG-I poly-ubiquitination mainly relies on Riplet activity (Figure 3) [138]. Cadena et al. showed that Riplet plays both ubiquitin-dependent and independent roles in RIG-I activation, depending on dsRNAs’ length. It mediates RIG-I poly-ubiquitination upon the recognition of short dsRNAs and induces RIG-I aggregate formation on longer dsRNAs, therefore amplifying RIG-I signaling [138].

TRIM25 catalytic activity requires its own RING domain dimerization [139], thus allowing TRIM25 PRY-SPRY domain to recognize the first RIG-I CARD and mediate the poly-ubiquitination at K172 on the second one (Figure 3) [106,140]. With less efficiency, RIG-I could also be activated by free K63 chains of ubiquitins synthetized by TRIM25 [14,141], and some studies suggest that these chains coil around CARDs and are then used as a scaffold for the next step [14,141,142].

RIG-I poly-ubiquitination leads to its tetramerization and TRIM25 release [14,133,143,144]. RIG-I then accumulates around mitochondria, where it binds to the mitochondrial antiviral signaling protein (MAVS) by its CARDs, leading to MAVS multimerization in filamentous structures [76,145,146,147,148]. MAVS mitochondrial localization is a determinant for innate immune response [149], and the MAVS C-terminal transmembrane domain allows its anchor into the mitochondrial membrane, where it aggregates upon RIG-I activation [146,150]. Through signaling cascades notably involving TAK1 and TRAFs proteins, MAVS multimerization leads to TBK1 and IKK kinases’ recruitment. In turn, TBK1 phosphorylates and activates IRF3 and IRF7 transcription factors that direct type I and type III IFNs expression, while IKK kinases mediate phosphorylation of NF-κB inhibitor IκB [14,97,108,145,146,151,152,153,154,155,156]. Once phosphorylated, IκB is targeted by the SCF^β-TrCP^ E3 ligase complex for proteasomal degradation, therefore activating NF-κB that translocates into the nucleus to direct pro-inflammatory cytokines as well as IFNβ expression [157,158,159]. Notably, it was shown that TBK1 and IKK kinases’ recruitment relies on K63 poly-ubiquitination mechanisms (Figure 2) [160,161].

### 3.2. ISGs Activation

Produced IFNs are secreted in an autocrine and a paracrine way from the infected cell and bind to IFNAR1/2 (type I IFNs) or IFNLR1 and IL10RB (type III IFNs) membrane receptors. Type I and type III IFNs are induced by the same PRRs and trigger similar downstream signaling, but recent studies suggest that type III IFNs’ response is the primary defense in epithelial cells, while type I IFNs’ machinery is more systemic and forms the second line of defense in case of broader infection [162]. Moreover, type III IFNs seem to trigger less inflammation than type I IFNs, thereby protecting epithelial tissue from immunopathology [163]. The binding of IFNs to their receptors activates signaling cascades that establish an antiviral state through phosphorylation of signal transducer and activator of transcription (STAT) 1 and STAT2 transcription factors by Janus kinase 1 (JAK1) and tyrosine kinase 2 (TYK2) [164,165]. These activated factors then associate with IRF9 to form an ISG factor 3 (ISGF3) complex that translocates into the nucleus where it binds to IFN-stimulated response element (ISRE) sequences in ISG promoters to induce their transcription (Figure 2) [164,166].

Many ISGs particularly inhibit IAV replication, including myxovirus resistance (MxA), IFN-induced transmembrane (IFITM), 2′–5′ oligoadenylate synthetase (OAS) and ribonuclease L (RNAseL) from OAS-RNase L pathway, protein kinase R (PKR), and ISG20 [97,167,168,169]. Human cytosolic protein MxA is a guanylate-binding protein (GBP) that inhibits IAV by targeting viral transcription and by binding to the NP protein [170,171,172]. The IFITM protein family contains three members (IFITM1, IFITM2 and IFITM3) that have antiviral activities by blocking the fusion of the viral particle with cellular host membrane at the uncoating step [169,173,174,175]. IFITM3 is notably down-regulated by its ubiquitination triggered by NEDD4 E3 ligase. In the absence of this E3 enzyme, cells are more resistant to IAV infection [76,77]. dsRNAs activate OAS and lead to the production of poly(A) chains that in turn bind to and activate RNAseL [176]. This ribonuclease then cleaves viral RNA and therefore inhibits viral replication in infected cells [177]. Moreover, products from viral RNA degradation can activate RIG-I, therefore amplifying IFN response [124,178]. The presence of dsRNAs in infected cells also activates the multifunctional PKR protein, which, for instance, phosphorylates the α subunit of the eukaryotic initiation factor 2 (eIF2α), subsequently inhibiting cellular but also viral proteins translation [179]. PKR also phosphorylates the NF-κB inhibitor IκB, and activates IRF3, leading to these transcription factors’ activation [180,181]. In addition, PKR stabilizes type I IFNs mRNA, thus enhancing IFNs production [182]. Finally, ISG20 is a 3′—5′ exonuclease known to inhibit several ssRNA viruses. During IAV infection, ISG20 binds to NP to block virus replication and transcription [168].

### 3.3. Inflammasome Complexes

In response to PAMPs, RIG-I also activates inflammasomes, multiproteic complexes consisting of nucleotide oligomerization domain (NOD)-like receptor family member LRR- and pyrin domain containing-3 (NLRP3), apoptosis-associated speck-like containing a caspase-recruitment domain (ASC) and pro-caspase-1 proteins [183,184]. Inflammasomes are involved in the defense against several viruses of IAV [183,185,186,187]. Indeed, NLRP3 is notably expressed in human bronchial epithelial cells [188]. Inflammasomes’ activation also relies on PAMP detection by NLRP3 and on PKR activity [189], as well as on protein flux through the M2 ion channel in the trans-Golgi network [190] and on IAV PB1-F2’s presence in the cell [191]. Inflammasomes stimulate caspase-1 activation, which in turn cleaves inactive pro-interleukin (IL)-1β and pro-IL-18 into mature IL-1β and IL-18 forms, respectively, which are then secreted to stimulate inflammatory response (Figure 2) [97,183,192].

### 3.4. Autophagy and Apoptosis

Autophagy, a self-eating mechanism, is involved in several cellular processes such as cell death or protein and organelle elimination through lysosomes, but it also plays a crucial part in innate immune response through TLRs activation, in inflammatory disorders, and in tumor suppression [193,194,195]. Depending on the host and on the virus strain, autophagy can further be used as a replication and pathogenesis enhancer [72,194,195,196,197,198] or is subverted by viral proteins such as IAV M2 to enhance virion stability [199,200]. Another cellular defense pathway triggered in response to intracellular pathogens is apoptosis, which is also involved in cell growth control [201]. This programmed cell death is notably hijacked by IAV to facilitate its replication [202]. Autophagy and apoptosis are both regulated by PKR signaling pathway [203,204].

## 4. The Innate Immune Battle between UPS and IAV

### 4.1. Antiviral Roles of UPS Factors

In addition to TRIM25 and Riplet, other UPS and UPS-like factors are involved in the antiviral pathway during IAV infection, either by directly and indirectly targeting viral proteins or by amplifying immune response.

TRIM and DUB proteins are notable key players in antiviral responses against IAV infection. Among TRIMs, TRIM5, TRIM14, TRIM22, TRIM32 and TRIM41 directly target viral proteins to restrain replication [105,116]. TRIM14 and TRIM41 bind to NP, leading to its ubiquitination and degradation, thus inhibiting vRNPs’ formation and viral RNA replication [205,206]. TRIM22 also interacts with NP to promote its K48-linked poly-ubiquitination and subsequent proteasomal degradation [71]. Nonetheless, it seems that some H1N1 strains evolved to evade TRIM22-mediated antiviral activity [207]. PB1 is targeted by TRIM32, which leads to a consecutive viral polymerase activity decrease [72]. Moreover, TRIM32 ligase activity stimulates the unc-51-like autophagy activating kinase 1 (ULK1), which controls autophagy induction through the formation of Beclin 1 complexes [208]. Several other TRIMs are implied in autophagy-associated antiviral mechanisms as well as in IAV-induced autophagy (reviewed in [105]). Another TRIM, TRIM44, stabilizes MAVS by inhibiting its proteasomal degradation, therefore stabilizing and enhancing the IFN response [209], while the TRIM28 protein is involved in pro-inflammatory cytokines production [210].

Among DUBs regulating the RIG-I dependent innate immune response, USP21 is able to remove RIG-I K63-linked poly-ubiquitin chains [211], USP4 hydrolyses RIG-I K48-linked chains [212], and OTU domain-containing protein 1 (OTUD1) stabilizes a not yet identified E3 ligase, thus facilitating MAVS degradation [213]. Interestingly, Otubain-1 DUB (OTUB1) was recently described as an antiviral key regulator in IAV-infected cells. Upon infection, OTUB1 is induced by type I IFNs and is associated with RIG-I at the mitochondrial membranes, where it activates RIG-I by a double mechanism [214]. First, OTUB1 specifically hydrolyses RIG-I K48-linked chains, and then it inhibits these RIG-I K48-linked poly-ubiquitinations through the formation of an E2-repressive complex [214,215,216].

Inflammasome complexes are also regulated by E3 and DUB enzymes. Indeed, the linear ubiquitin chain assembly complex (LUBAC) and TRIM33 E3 ligases promote inflammasomes’ formation [217,218], while breast cancer 1 (BRCA1)/breast cancer 2 (BRCA2)-containing complex subunit 3 (BRCC3) de-ubiquitinates NLRP3 [219].

Other mechanisms, such as SUMOylation, ISGylation and NEDDylation, also regulate several viruses, such as IAV [220,221,222]. SUMOylated TRIM28 acts with other proteins to inhibit endogenous retroviral (ERV) elements’ expression [223,224,225,226,227,228]. Cellular ERVs can be recognized by RIG-I and/or MDA5, therefore leading to aberrant IFN stimulation [229,230,231,232,233,234]. In IAV-infected cells, Schmidt et al. observed a diminution of SUMOylated TRIM28 levels, thus preparing cells for antiviral state establishment [220]. It was also shown that ISG15 conjugation, even at a low level, has a negative impact on the ability of the virus to replicate [235,236]. In addition, MDM2 E3 ligase seems to enhance M1 and PB2 NEDDylation, leading to the destabilization of these viral proteins and the inhibition of IAV replication in vitro and in vivo [222,237]. On the contrary, Sun et al. observed that IAV infection leads to the activation of NEDDylation pathway, which promotes virus growth but also pathogenicity. Indeed, cullin 1 NEDDylation can activate the NF-κB pathway and lead to the over-expression of pro-inflammatory cytokines that enhance viral pathogenicity [238]. NEDDylation modifications therefore seem to play a balancing role in the regulation of IAV replication.

Finally, the UPS is also indirectly involved in defense against IAV, through enhancement of proteasomal-mediated degradation of viral proteins. For example, Cyclophilin A accelerates the degradation of M1 [239], while the long isoform of zinc finger antiviral protein (ZAP-L) associates with PA and PB2 to promote their presumably K48-linked poly-ubiquitination, although ZAP-L activity is counteracted by PB1 protein [75].

### 4.2. IAV Hijacks UPS to Evade Innate Immune Response

As a response to antiviral state triggered by infected cells, many viruses evolved to counteract RIG-I-mediated innate immune response [240,241]. For instance, IAV PB1-F2 and PB2 proteins interact with MAVS, NP prevents PKR-mediated IRF3 phosphorylation by binding to HSP40, and HA expression leads to IFNAR1 ubiquitination and degradation, thus inhibiting type I IFN production and response pathways [242,243,244,245]. Accordingly, IAV can use the UPS not only to facilitate its replication, but also to evade the host’s immune system.

Hence, different examples of UPS hijacking by IAV are described in the literature. Among others, the ubiquitination of M2 on its K78 residue regulates IAV replication by coordinating apoptosis and autophagy, two mechanisms used by the virus to spread and to control infection-mediated cellular death, even though the involved E3 ligase is still unknown [246]. Regardless of these significant examples, the main IAV virulence protein remains NS1, acting by direct interactions with proteins such as TRIM25 or indirect interactions by regulating host gene expression [107,247,248,249,250,251,252,253,254,255].

## 5. Mechanisms Developed by NS1 to Inhibit the Innate Immune Response

### 5.1. IAV NS1 Protein

The RNA segment 8 of the IAV genome encodes for a mRNA, from which is synthetized the non-structural protein NS1. An alternative splicing of this mRNA results in the production of the NEP protein, which represents 10–15% of the segment 8 transcripts [256,257]. A third transcript with a punctual mutation and a subsequently alternative splicing is also found in some viral strains, therefore expressing a NS3 protein with the same terminal ends as NS1 [258]. NS1 is generally a 230 amino acid (AAs) protein, even though its length may vary due to mutations depleting the stop codon at position 231 or creating a premature stop codon. For example, human IAV NS1 displayed 237 AAs from the late 1940s to the middle of the 1980s, while NS1 from the 2009 pandemic H1N1 strain only harbors 219 AAs [252,259,260]. 

NS1 possesses an RNA-binding domain (RBD) in N-terminus (AAs 1–73), separated by a flexible linker region from an effector domain (ED) (AAs 85–207), which is followed by a C-terminal tail. The RBD forms three α-helices that are crucial for NS1 dimerization and for interaction with dsRNA [261,262,263]. Amino acids at position 38 and 41 are important for this binding function [263,264], while dimerization allows NS1 homodimer and then oligomer formation in infected cells [265,266,267,268]. The RBD also displays a nuclear localization signal (NLS) from AA 35 to 41 [269,270]. The ED contains a β-sheet structure and a long central helix as well as a nuclear export signal (NES) (AAs 137–146). Together with NLS, this signal allows NS1 to shuttle from the nucleus to the cytoplasm during infection [271]. Interestingly, several cellular proteins have been shown to interact with NS1 ED [272]. Similarly to the linker region (10 to 15 AAs long), the C-terminal tail varies in length (11 to 33 AAs long), and it may contain, depending on the viral strain, a PDZ-binding motif involved in viral pathogenesis [273,274]. Notably, H1N1 1918 NS1 contains a SUMOylation consensus site embedded in the PDZ-binding domain [275]. Most of the IAV strains also display a NLS in the tail domain (AAs 216–221) (Figure 3) [269].

### 5.2. NS1 against Host Antiviral Response

NS1 is known to inhibit the antiviral response in the host cell by means of a wide range of mechanisms, including through protein interactions, host shutoff and ubiquitination perturbations (reviewed in [252,272,276]), turning NS1 into a key protein for viral replication cycle. Some studies therefore suggest that IFNβ-competent cells infected with IAV lacking NS1 considerably impair viral replication [277,278], as well as cells infected with IAV expressing a truncated form, low level or functional mutations of NS1 [279,280,281,282]. Non-canonical antiviral pathways seem to be also targeted by NS1. Indeed, Schmidt et al. showed that NS1 can bind to ERV dsRNA and then inhibits the subsequent innate immune response [220].

#### 5.2.1. IFNs, Cytokines and ISGs Inhibition by NS1

During the battle between IAV infection and innate immune response, NS1 inhibits IFN activation through the interaction with proteins such as Riplet [283] and IRF3 [247]. Moreover, it was shown that the NS1 ED of some IAV strains suppresses TRAF3 K63-linked poly-ubiquitination, probably through DUBs recruitment. This leads to the disruption of the MAVS and TRAF3 complex and to the impairment of subsequent IRF3 phosphorylation [284]. NS1 also inhibits NF-κB activation by blocking the β subunit of IKK [285]. Its RBD seems to play a crucial role in cytokine production and in cytokine sensitivity [286].

In addition, NS1 was shown to regulate the expression of some JAK/STAT signaling inhibitors by binding to the DNA methyltransferase 3B (DNMT3B). Upon IAV infection, NS1 dissociates DNMT3B from gene promoters and changes its localization to the cytosol, where it is K48-linked poly-ubiquitinated and degraded, therefore leading to an enhanced expression of JAK/STAT signaling inhibitors [287]. NS1 also inhibits some specific ISGs, such as IFITM3, the expression of which is attenuated by NS1-mediated eIF4B degradation [272]. By binding viral RNA, NS1 competes with cellular RNA, binding proteins such as OAS, RNAse L, PKR and RIG-I, thus protecting them from degradation and inhibiting antiviral response (Figure 2) [251,288,289,290].

NS1 antagonizes inflammasome by a direct interaction with NLRP3, which prevents ASC ubiquitination and inflammasome activation (Figure 2) [185,291,292].

Finally, NS1 up-regulates the phosphoinositide-3-kinase (PI3K) pathway through direct interaction with the regulating subunit p85β of PI3K, causing downstream phosphorylation of the protein kinase B (Akt) [293,294,295,296]. Notably, p85β possesses a CCD to which NS1 ED binds [297,298]. This leads to an increase in the viral internalization rate, apoptosis inhibition [299], and an enhancement of IRF3 activity [300,301]. While NS1 seems to inhibit apoptosis, recent studies suggest that it also has a pro-apoptotic effect [302]. NS1 dual function in apoptosis regulation therefore may vary depending on the stage of viral infection. Indeed, induction of apoptosis seems to be essential for vRNPs’ nuclear trafficking at the beginning of the infection, while the limitation of apoptosis at later stages could prevent premature viral particles release and death of infected cells [303].

#### 5.2.2. Host Shutoff

Another mechanism triggered by NS1 to impair the antiviral state in infected cells is host shutoff, i.e., inhibition of host genes expression. NS1 mainly inhibits cellular mRNAs processing by binding to CPSF30, a key component of pre-mRNA 3′-end formation machinery [260,279,304,305,306,307]. This interaction prevents the recognition by CPSF30 of poly(A) signals at 3′-end of mRNAs during their transcription, thereby blocking immature mRNAs’ cleavage and poly(A) tail addition [307,308]. Poly(A) tail allows mRNAs’ stabilization, nuclear export, and translation [308]. In its absence, immature mRNAs thus accumulate in the nucleus of infected cells, leading to a general host shutoff, affecting, among other things, the expression of IFNs, ISGs, and pro-inflammatory proteins [272,306,309]. In this way, NS1 also binds and inhibits PABPII, the nuclear poly(A)-binding protein II, which stimulates the synthesis of long poly(A) tails [309]. It was shown that NS1 RBD is able to bind host dsDNAs, therefore inhibiting the loading of transcriptional machinery to IAV antiviral genes [310]. Nuclear RNA export factor 1 (NFX1), ribonucleic acid export 1 (RAE1) and p15 are examples of mRNA export machinery proteins also targeted by NS1 from the H1N1 WSN strain [311]. Additionally, NS1 enhances viral mRNAs translation to the detriment of cellular mRNAs by recruiting eIF4GI to the viral mRNA 5′-UTR region [312].

### 5.3. Ubiquitination Perturbations by NS1

#### 5.3.1. TRIM25

One of the main antiviral targets of NS1 is TRIM25 [107,283], therefore interfering with its E3 ligase activity and with IFN signaling pathway activation (Figure 2) [107]. The interaction between PRY-SPRY domain of TRIM25 and its own CCD is essential to mediate RIG-I ubiquitination [118], and NS1 directly competes with this binding [107,118,144]. Indeed, NS1 presumably binds to TRIM25 CCD via its functional ED amino acids 96 and 97 (Figure 3) [107]. Interestingly, this interface is the same as the one in NS1 and p85β interaction, suggesting that NS1 uses the same ED surface to recognize similar structures in various targets [118,297,298]. NS1 and TRIM25 complex formation therefore inhibits the correct juxtaposition of TRIM25 PRY-SPRY and RING domains, and thus TRIM25 multimerization and activity [107,118]. However, this binding does not impair free K63-linked poly-ubiquitin chains formation by TRIM25, neither the formation of TRIM25 and RIG-I complex in infected-cells cytoplasm [118]. On the contrary, NS1 also interacts with RIG-I and inhibits its activation by sequestrating its RNA helicase and its ligand [255,313,314]. Moreover, a recent study suggested that NS1 RBD from 1918 H1N1 strain directly binds to the RIG-I second CARD and inhibits its ubiquitination (Figure 3) [315], and that the R21Q natural mutation of some NS1 proteins intriguingly impairs this inhibition, leading to a stronger immune response [315,316]. This mutation could therefore be a specific adaptation of some strains to host species [283,315]. Indeed, it has been shown that H3N2 and H2N2 IAV strains infections lead to a high IRF3 and IFNβ activation, despite a proper interaction between NS1 and TRIM25 [317,318]. Meyerson et al. thus suggested that NS1 inhibits another antiviral function of TRIM25, independent of its ligase activity and of RIG-I. In fact, TRIM25 can bind to vRNP RNA and proteins in infected-cell nucleus, preventing viral RNA elongation [196]. Finally, NS1 is also able to suppress TRIM25 expression at a transcriptional level through up-regulation of Dot1L methyltransferase during infection [319].

#### 5.3.2. DUBs

NS1 inhibits the RIG-I pathway by up-regulating the expression of A20 DUB (other name TNFAIP3), a negative regulator of the innate immune response [99,320,321]. A20 suppresses the antiviral state established during infection [322], and its up-regulation by NS1 seems to be proportional to strain virulence [99]. A20 is a dual-ubiquitin editing enzyme that contains a N-terminal DUB activity and seven Zn finger domains with E3 ligase activity in its C-terminal region, which regulate innate immune response [323,324].

OTUB1 is also targeted by NS1, even though the mechanism is still undefined. NS1 could trigger OTUB1 proteasomal degradation at the later stages of infection, leading to the inhibition of IRF3 and NF-κB activation, and thus preventing the antiviral response (Figure 2) [214].

#### 5.3.3. p53 and MDM2

p53 is a well-known transcription factor and tumor suppressor that accumulates in the nucleus upon cellular stress [325,326]. It regulates several biological processes, including apoptosis and antiviral mechanisms by regulating infected-cell fate [327,328,329,330,331,332]. p53 activates, among other things, the transcription of its main negative regulator, MDM2, a RING E3 ligase that in turn binds to p53 and mediates its poly-ubiquitination and proteasomal degradation. MDM2 can also mediate p53 NEDDylation [60]. p53 stability is therefore dependent on its interaction with MDM2, as well as on the MDM2 level or cellular localization [295].

IAV was shown to regulate p53 transcriptional activity, through its stabilization by NS1 [330,333,334]. This p53 stabilization in infected cells is associated with a defect of MDM2-mediated ubiquitination of p53 [335]. Pizzorno et al. showed that NS1 is responsible for proteasomal-mediated degradation of MDM2 at early stages of infection, and that it also modifies MDM2 subcellular localization during IAV infection [336]. As with p53, MDM2 could also plays an antiviral role, independently of its ligase activity and p53 [336]. According to the ambivalent role of apoptosis during IAV infection, MDM2 could be used by NS1 at early stages of infection to promote IAV propagation [303]. However, MDM2 is involved in many other pathways linked to IAV infection, such as cell cycle control or NF-κB signalization, perhaps ensuring antiviral effect by other mechanisms [337,338].

#### 5.3.4. SUMOylation System

Similarly to NP discussed previously, NS1 also interacts with the cellular SUMOylation system during IAV infection. Indeed, some studies suggest that most of IAV strains possess a SUMOylated NS1 protein, and that different lysine residues can be targeted, even though K131 seems to be the main NS1 SUMOylation site. These studies also highlight the crucial, but not essential, role of NS1 SUMOylation in virus replication [339,340,341]. Santos et al. notably showed that SUMOylation at K70 and K219 has no impact on NS1 stability and localization but can affect NS1 oligomerization and therefore NS1 functions. Furthermore, they showed that a modification of NS1 SUMOylation level negatively impacts the ability of NS1 to counteract IFN response [340].

Moreover, SUMOylation of NS1 was recently shown to take part in host shutoff by enhancing NS1 association with nuclear ribonucleoprotein complexes that control the activity of the RNA polymerase II [275].

#### 5.3.5. ISG15

Interestingly, NS1 from influenza B (IBV) virus also antagonizes the RIG-I-mediated antiviral response [342]. Notably, IBV NS1 binds to the N-terminal ubiquitin-like domain and to the linker region of ISG15 [46,343,344], inhibiting its activity and its secretion [39]. Moreover, IBV NS1 binds and sequesters ISGylated proteins, such as NP, to prevent its undesirable and premature inclusion into vRNPs [235,343].

## 6. Conclusions

The ubiquitin-proteasome system, as well as ubiquitin-like machineries, are key cellular processes involved in various pathways, including notably innate immune response. Upon influenza A virus infection, ubiquitination-mediated RIG-I activation leads to an antiviral state establishment in infected and surrounding cells, thus inhibiting IAV replication through IFNs, ISGs and pro-inflammatory cytokines expression. Several UPS factors are described to enhance and stabilize this innate immune response and to directly target IAV proteins for their proteasomal-mediated degradation. In turn, IAV evolved to hijack the UPS and use it to enter the cell and spread, disrupting UPS activity from antiviral to proviral roles. Furthermore, the viral non-structural protein NS1, a well-known virulence factor, developed many mechanisms to antagonize innate immunity, from direct interactions with cellular proteins to the general inhibition of host gene expression. Interestingly, NS1 specifically targets UPS factors such as TRIM25, A20, OTUB1 and MDM2, SUMOylation system, as well as ISG15 in the case of influenza B virus. NS1 activity seems to vary depending on the infection timing, and further investigations are therefore needed to decipher the role of NS1 in ubiquitin perturbations during the course of IAV infection. We are currently investigating the identification of UPS factors interacting with IAV NS1 and the functional characterization of some of them.

## Figures and Tables

**Figure 1 viruses-13-02309-f001:**
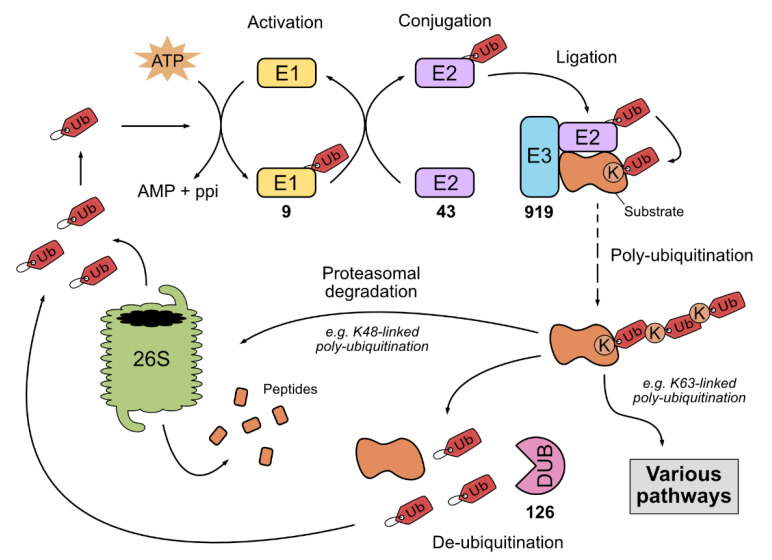
Ubiquitin-proteasome system. Ubiquitin (Ub) is first attached to the ubiquitin-activating enzyme E1 in an ATP-dependent way. Ubiquitin is then transferred to the ubiquitin-conjugating enzyme, E2. An isopeptide bound is finally catalyzed between the ubiquitin and a lysine (K) residue on a protein substrate by the ubiquitin ligase, E3. The ubiquitinated protein can subsequently be poly-ubiquitinated through K residues within ubiquitin itself. Depending on the linkage type, the protein will then undergo different fates in various cellular pathways. De-ubiquitinase enzymes (DUB) can recognize and hydrolyze isopeptide bonds formed between ubiquitin and protein substrate and between ubiquitin moieties, leading to the recycling of ubiquitin. K48-linked poly-ubiquitination leads to targeted protein degradation by the 26S proteasome, releasing free ubiquitins, which are reincorporated in the loop or used in other pathways. For each category of enzymes (E1, E2, E3 and DUB), the number of proteins in humans are indicated (source: IUUCD 2.0 database).

**Figure 2 viruses-13-02309-f002:**
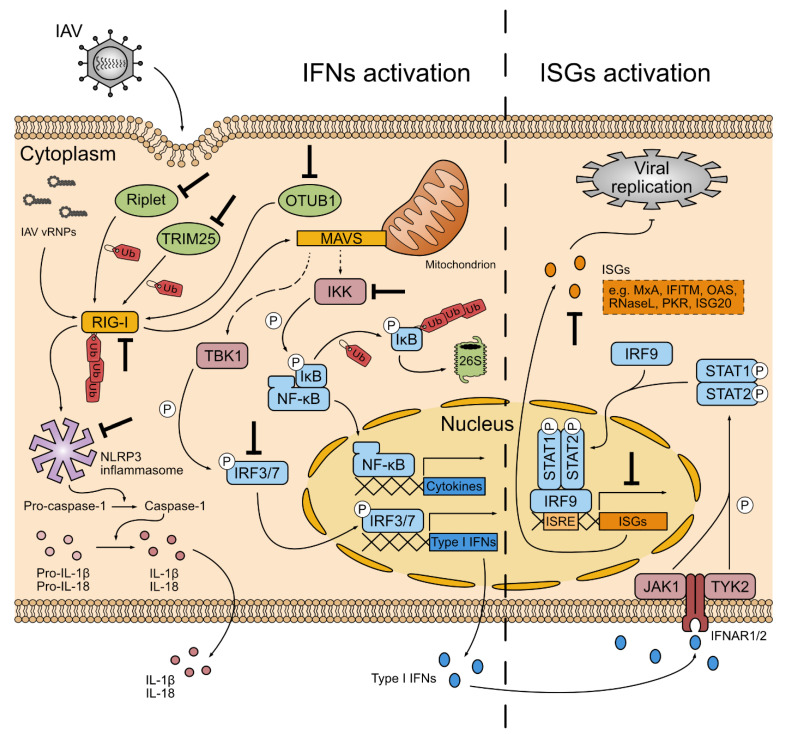
Innate immune response during influenza A virus infection and NS1 hijackings. IFNs activation (left): viral particles are incorporated in the host cell through endosomes formation, leading to vRNPs’ release in the cytoplasm. The RLR sensor RIG-I is then activated by their presence and is K63 poly-ubiquitinated by Riplet and by TRIM25. The DUB OTUB1 also activates RIG-I through inhibition of its K48-linked poly-ubiquitination. In turn, RIG-I activates the NLRP3 inflammasome, that stimulates caspase-1 activation. Caspase-1 then mediates the maturation of interleukins IL-1β and IL-18, which are then secreted to promote inflammatory response. After its activation, RIG-I mainly binds to MAVS and promotes its multimerization at the mitochondrial membrane. Through signaling cascades not depicted here, MAVS mediates the recruitment of TBK1 and IKK kinases, which phosphorylate IRF3 and IRF7 transcription factors, and the NF-κB inhibitor IκB, respectively. IκB is then poly-ubiquitinated and degraded by the 26S proteasome. Activated IRF3 and IRF7, as well as activated NF-κB then translocate into the nucleus to promote type I IFNs (IFNα et IFNβ) expression, and pro-inflammatory cytokines, respectively. Notably, NF-κB also mediates IFNβ expression. ISGs activation (right): type I IFNs are then secreted in an autocrine and a paracrine way and bind to IFNAR1/2 heterodimeric receptor. Upon IFN binding, JAK1 and TYK2 kinases are activated and mediate phosphorylation of STAT1 and STAT2 transcription factors. These proteins then translocate into the nucleus with IRF9 and bind to ISRE sequences in ISG promoters to induce their transcription. Several ISGs are known to particularly counteract IAV replication (examples in the orange box). In turn, the NS1 protein hijacks innate immune response by targeting different proteins (bold black arrows), and notably ubiquitin factors (green).

**Figure 3 viruses-13-02309-f003:**
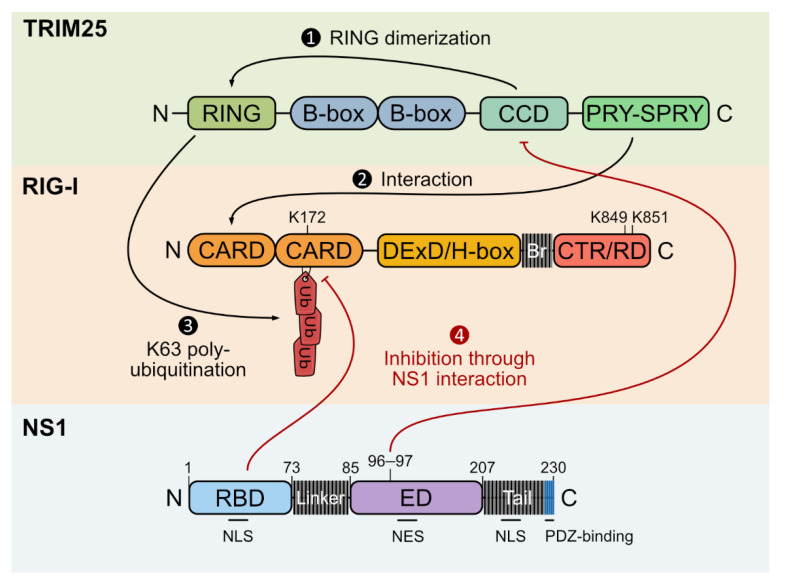
Schematic representation of TRIM25, RIG-I and NS1 protein structures and interactions. Upon infection, TRIM25 CCD first mediates RING domain dimerization (1) and subsequent complete TRIM25 dimerization, allowing interaction between TRIM25 PRY-SPRY domain and RIG-I first CARD (2). TRIM25 RING domain then mediates K63-linked poly-ubiquitination of the lysine 172 (K172) on RIG-I second CARD (3), leading to antiviral response activation. In turn, IAV NS1 protein can inhibit these two proteins (red arrows) (4). NS1 RBD from 1918 H1N1 strain directly binds to and inhibits poly-ubiquitination of RIG-I second CARD, while amino acids 96 and 97 in ED interact with TRIM25 CCD, therefore inhibiting TRIM25 dimerization.

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
