# Peer review of "How Influenza A Virus NS1 Deals with the Ubiquitin System to Evade Innate Immunity"

_viruses, 2021, doi:10.3390/v13112309_

Round 1
Reviewer 1 Report
Comments for the authors of the Viruses Review Manuscript:
The authors of the Viruses manuscript “How influenza A virus NS1 deals with the ubiquitin system to evade innate immunity”, present a review of the literature that focuses on host:pathogen interactions related to influenza virus infection and the ubiquitin system. Their presentation of the ubiquitin system, viral infection, type I interferon responses, and influenza NS1 in the context of intracellular interactions is well-written and a good resource for information on this topic. Their description of intracellular interactions that the host uses to inhibit viral replication and that the virus uses to avoid host recognition help bring together both sides of this complex host:pathogen interaction. I greatly appreciated the use of Figures throughout the document to illustrate the mechanisms presented, and the description of these interactions from the view of both the host and the pathogen. I found this review to be an interesting read that brings together references for researchers that are new to the field, as well as those that are familiar with this research area. I have only a few comments that I would like the authors to consider as they move through the revision process.
General Comments:
- There were multiple errors on sentence structure and grammar that should be addressed during revision. A couple that I would like to point out are sentence fragments on lines 297-298 and 350-351.
Author Response
We thank the reviewer for his positive comments on our review. We tried to correct the english mistakes as best could and hope the reviewer will consider the text is improved.
Reviewer 2 Report
The review focused on the interaction of NS1 protein of influenza A virus (IAV) with the ubiquitin system for innate immunity evasion. Since NS1 protein is an important protein to inhibit host innate immunity, and ubiquitination play an important role in triggering the host innate immunity, it is worthy to discuss the relationship between NS1 protein and the ubiquitin system. However, the organization and contents need to be improved.
- Usually, there is an introduction to explain the importance for writing the review.
- In the ubiquitin & ubiquitin-like systems section, the paragraph of “Focus on the TRIM proteins” may move and combine to section 4 “The innate immune battle between UPS and IAV”. Also, the NEDDylation should be stated in brief.
- The numbers of proteins E1, E2, E3, DUB in human were not necessary to labeled in figure 1, since the numbers were not accurate.
- The Figure 2 didn’t include the other important NS1 interaction protein, which related to ubiquitination, such as TRAF3 and DNMT3B,. (Qian W et al. Front. Immunol. 2017, 8:779. Liu S et al. J. Virol. 2019, 93:e01587-18.). Also the NLRP3 as the review mentioned was not shown in figure 2.
- The authors described too much background information about the IFN and cytokine activation, the UPS and IAV. The authors should focus on how influenza A virus NS1 deals with the ubiquitin system to evade the innate immunity.
- The authors should include the references for NS1and SUMOylation in the section 5.
Round 2
Reviewer 2 Report
The manuscript has been revised as suggestions.